# Lamellar Orientation of a Block Copolymer via an Electron-Beam Induced Polarity Switch in a Nitrophenyl Self-Assembled Monolayer or Si Etching Treatments

**Hiroki Yamamoto [1,*], Guy Dawson [2], Takahiro Kozawa [3] and Alex P. G. Robinson [2]** 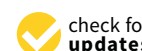

[1]  National Institutes for Quantum and Radiological Science and Technology, 1233 Watanuki-machi, Takasaki, Gunma 370-1292, Japan

[2]  School of Chemical Engineering, University of Birmingham, Edgbaston, Birmingham B15 2TT, UK; GDD015@student.bham.ac.uk (G.D.); a.p.g.robinson@bham.ac.uk (A.P.G.R.)

[3]  The Institute of Scientific and Industrial Research, Osaka University, 8-1 Mihogaoka, Ibaraki, Osaka 567-0047, Japan; kozawa@sanken.osaka-u.ac.jp

*   Correspondence: yamamoto.hiroki@qst.go.jp; Tel.: +81-27-346-9689

**Abstract:** Directed self-assembly (DSA) was investigated on self-assembled monolayers (SAMs) chemically modified by electron beam (EB) irradiation, which is composed of 6-(4-nitrophenoxy) hexane-1-thiol (NPHT). Irradiating a NPHT by EB could successfully induce the orientation and selective patterning of block copolymer domains. We clarified that spatially-selective lamellar orientations of polystyrene-*block*-poly(methyl methacrylate) (PS-*b*-PMMA) could be achieved by a change of an underlying SAM. The change of an underlying SAM is composed of the transition of an $NO_2$ group to an $NH_2$ group, which is induced by EB. The modification in the polarity of different regions of the SAM with EB lithography controlled the lamellar orientation of PS-*b*-PMMA. The reduction of the NPHT SAM plays an important role in the orientation of block copolymer. This method might significantly simplify block copolymer DSA processes when it is compared to the conventional DSA process. By investigating the lamellae orientation with EB, it is clarified that only suitable annealing temperatures and irradiation doses lead to the vertical orientation. We also fabricated pre-patterned Si substrates by EB lithographic patterning and reactive ion etching (RIE). DSA onto such pre-patterned Si substrates was proven to be successful for subdivision of the lithographic patterns into line and space patterns.

**Keywords:** electron beam; directed self-assembly; block copolymer; self-assembled monolayers; lithography; polystyrene-*block*-poly(methyl methacrylate)

## 1. Introduction

With further miniaturization of devices, sub-10 nm feature sizes are anticipated which will not easily be obtained by extreme ultraviolet (EUV) or electron beam (EB) lithography techniques in mass production, due to patterning limitations, tool costs, or low throughput and so on. The fusion of top-down and bottom-up approaches to nano-patterning has attracted significant attention from researchers because conventional top-down lithographic techniques are now approaching such fundamental limitations. Self-assembly of block copolymers enables the fabrication of features of less than 20 nm without using an expensive exposure tool. In particular, the directed self-assembly (DSA) of block copolymers has attracted significant attention as a promising nanofabrication technique to exceed the fundamental limitations of top-down lithography. Block copolymers have the potential to

be used in semiconductor manufacturing [1–6] and DSA is currently being considered for future nodes of the Interational Technology Roadmap for Semiconductors (ITRS) [7].

Up to now, a lot of effort has been devoted to controlling the self-assembly of block copolymers using the techniques of grapho-epitaxy [2], [8–11] and chemo-epitaxy [12,13]. Generally, the orientation of the block copolymer can be controlled by precisely tuning the chemistry of the interface between the block copolymer and the substrate. The basic method for achieving perpendicular orientation of the block copolymer is to balance the interfacial interactions of each block of the block copolymer with the substrate. Neutral surfaces have been shown to induce a perpendicular domain orientation in block copolymer thin films [14,15]. This strategy utilizes the inherent versatility of random copolymers, which allows the surface energy or surface characteristics to be tuned due to the change in chemical composition of the random copolymer. A more general approach to controlling interfacial and surface interactions using a crosslinkable random copolymer thin film has been developed [16]. In addition, there are several methods such as solvent annealing [17], the manipulation of rough substrates [18,19], chemical modification of underlying substrates [15,20,21] and so on to control the orientation of domains in block copolymer thin films. In particular, surfaces with neutral wettability to the polystyrene-*block*-poly(methyl methacrylate) (PS-*b*-PMMA) block copolymers, such as self-assembled monolayer (SAM) modified surfaces or random styrene-methacrylate copolymer films (PS-*r*-PMMA) have been studied to induce perpendicular orientation for the self-assembly of PS-*b*-PMMA. Pre-patterned surfaces, which consist of alternating neutral and preferential surfaces, have been accomplished by the selective oxidation of a neutral SAM with resist mask [16,18,19]. However, current methods for achieving the pattering of underlying substrates are generally composed of a large number of steps. In order to simplify and improve DSA processes, several strategies have been reported to control areas of block copolymer domain orientation using radiation. Approaches including a photodefinable substrate film [22], EB sensitive materials [23], x-ray sensitive self-assembled monolayers [15], the selective cross-linking of the surface of underlying substrates via UV light [24,25], and reactive ion etching [26–28] have all been reported. In addition, disordered block copolymers that order in response to light [29–32] and electrohydrodynamic jet printing [33] have been reported. However, a number of challenges remain with these approaches.

It is very valuable to control the surface chemistry of specific regions in order to locally change the orientation of block copolymer domains by lithography techniques. In particular, it is suitable for the modification of the substrate surface to use EB lithography and has the advantage of making it easily possible to fabricate less than 20 nm feature size patterns. In addition to high resolution, EB lithography makes it possible to fabricate a lot of patterns by adjusting the irradiation dose and beam position. One strategy to induce lamella to form perpendicular to the substrate with controlled orientation in the plane of the film is the nanopatterning of substrates with alternating regions that are wetted differently by the different blocks of copolymer. In particular, tuning the polarity of the SAM-covered surface can control the wetting behavior of block copolymer films. It has been reported that such a surface pattern could be transferred to poly(styrene-*b*-2-vinylpyridine) (P(S-*b*-2VP)) films by using strip patterns of SAMs of $CH_3$– and HO– terminated alkanethiols on gold substrate [34–36]. Furthermore, it has been reported that the wetting behavior of a polymer was controlled by the functionality, or surface chemistry of the SAMs such as strip patterns of $CH_3$– and COOH–terminated alkanethiols on gold [37,38], and the stripe patterns of gold and SAMs of $CH_3$–terminated alkanethiols on gold [39]. Moreover, the surface chemistry of the SAMs was modified upon exposure to x-rays in air to incorporate oxygen into polar function groups on the surface of the SAMs [40]. However, none of these SAMs have been directly patterned by EB reduction of the SAM to fabricate alternating regions that are wetted by the different blocks of copolymer. In fact, a patterned resist has been utilized as a mask during the etching process to create a chemical pattern on the buffer layer [26,27]. In addition to orientation control, photopatternable interfaces have been used to define trenches of DSA via grapho-epitaxy using photolithography [2] and to achieve DSA for chemo-epitaxy with lithography in tandem with other processes [13,41–45].

In this study, we demonstrate that lamella orientation of PS-*b*-PMMA films can be achieved using EB induced conversion of SAMs, such as the transition of a $NO_2$ group to an $NH_2$ group upon electron irradiation. Also, we fabricated pre-patterned Si substrates by EB lithographic patterning and reactive ion etching. DSA onto such pre-patterned Si substrates modified with PS-*r*-PMMA results in subdivision of the lithographic patterns into line and space patterns. Furthermore, we present experiments investigating the difference in the direction of a block copolymer thin film in contact with a chemically patterned underlying substrates and pre-patterned Si substrates.

## 2. Materials and Methods

Gold was deposited onto silicon substrates in a sputter coater (Edwards 306 auto, Edwards, West Sussex, UK), using an argon pressure of 1 Pa, and sputtering power 100 W, to a thickness of 100 nm, (measured on a sacrificial sample using a surface profiler (Dektak 3st, Veeco, New York, N.Y., USA). The gold coated samples were then cleaned in piranha solution (mixture solution of 30% hydrogen peroxide and concentrated sulfuric acid) (Warning: Piranha solution is very reactive and corrosive) for 5 min at room temperature. 6-(4-nitrophenoxy) hexane-1-thiol (NPHT) was prepared as detailed elsewhere [46]. To deposit the SAMs, a 10 mmol solution of NPHT was created in ethanol solvent, and the gold samples submerged in the solution for 48 h. Deposition was terminated using a 15 s rinse in the solvent and the samples were then dried with nitrogen. The rinse and dry was repeated twice to ensure maximum removal of physisorbed multilayers.

Samples were then irradiated with EB to produce a patterned SAM layer. They were patterned at exposure doses of 10 mC/cm$^2$ and 50 mC/cm$^2$. Lamella forming PS-*b*-PMMA (PS: 53 kg/mol, PMMA: 54 kg/mol) was purchased from Polymer Source, Inc. and used as received. Propylene glycol mono methyl ether acetate (PGMEA) was used as casting solvent. PS-*b*-PMMA block copolymer thin films were formed onto the patterned SAM layer coated substrates after EB irradiation. Subsequently, they were annealed at 150 °C, 190 °C, and 250 °C in a vacuum for a sufficiently long time. Then the films were also treated with an oxygen reactive ion etching (RIE) process to selectively remove the PMMA domains. After selective removal of PMMA, the remaining PS patterns on the substrate can be obtained. A tungsten was sputter-coated on the surface to prevent charging. The resulting morphology was recorded using a field emission-scanning electron microscope (FE-SEM S-5500, Hitachi-hitec, Japan) operated with an acceleration voltage of 1 kV.

Deep topographic features were also fabricated in silicon (Si) using EB lithography and plasma etching techniques. The surface of the lithographically defined templates was coated with PS-*r*-PMMA by spin-coating from a suitable solvent, and the samples were baked at 110 °C for 90 s. After baking, PS-*b*-PMMA block copolymer thin films were formed from PGMEA solutions onto the samples, which were then annealed at 190 °C for 24 h. After the self-assembly of block copolymer, the sample was subjected to an $O_2$ plasma to remove the PMMA domains. The resulting patterns were recorded using FE-SEM after tungsten coating.

## 3. Results and Discussion

Figure 1a shows a schematic of the typical lithographic patterning method used to chemically pattern a SAM surface, in order to subsequently induce directed self-assembly via chemo-epitaxy. The method typically proceeds by the patterning of SAMs or grafted polymer monolayers using lithographically defined resist patterns as a mask. However, these approaches require a number of complex processes such as resist coating, exposure, development, exposure in the presence of $O_2$, the difficulty of surface neutrality, and so on. Compared to the complex processes used for traditional chemo-epitaxy, a directly chemically patternable SAM would greatly simplify the process as shown in Figure 1b. Thus, the number of steps can be reduced in the DSA process. In addition, this method reduces the probability of defectivity, for instance caused by residual resist on the patterned SAM.

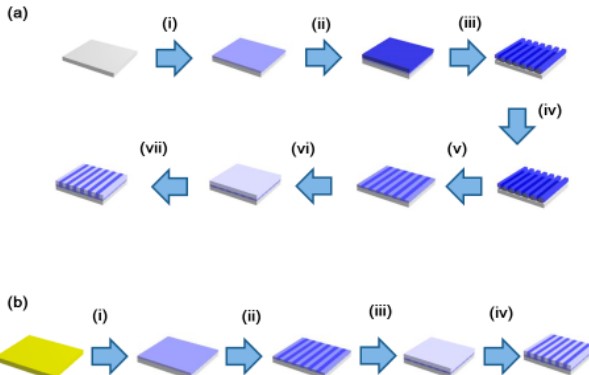

**Figure 1.** (**a**) Schematic presentation of block copolymer lamella orientation process using a conventional chemically patterned surface. (i) self-assembled monolayer (SAM) coating on Si. (ii) resist coating. (iii) Resist was exposed and developed. (iv) SAM was patterned in the presence of $O_2$ though resist mask (v) Resist was removed and surface neutrality. (vi) Block copolymer was spin-coated. (vii) Block copolymer was self-assembled to match underlying SAM patterns. (**b**) Schematic presentation of block copolymer lamella orientation process using a directly chemically patternable SAMs method. (i) SAM coating on Si. (ii) SAM was directly patterned by EB. (iii) Block copolymer was spin-coated. (iv) Lamella orientation of block copolymer was self-assembled in underlying SAM patterns.

EB patterning was chosen for controlling block copolymer features because EB lithography tools can pattern areas nearly as small as the domain of block copolymer. By exposing a responsive interfacial surface of SAMs to EB, selective orientation of block copolymer domain can be achieved. In other words, after EB irradiation the SAMs underwent selective reduction. It has previously been reported that EB lithography and the irradiation of x-ray can induce the transition of the SAM terminal function aromatic nitro ($NO_2$) moieties to aromatic amino ($NH_2$) moieties [46–53]. Thus, EB irradiation results in a chemically striped patterned substrate consisting of alternating stripes of SAM with an aromatic $NO_2$ terminal functionality and SAM with aromatic $NH_2$ terminal functionality, as shown Figure 2. Chemical analysis of substrate pre- and post-EB modification was not performed as the EB exposure tool's time and analytical tools were limited. However, X-ray photoelectron spectroscopy (XPS) data of the SAM substrates by pre- and post- x-ray modification [52] and XPS data of different SAM substrates by post- EB modification [49] are available.

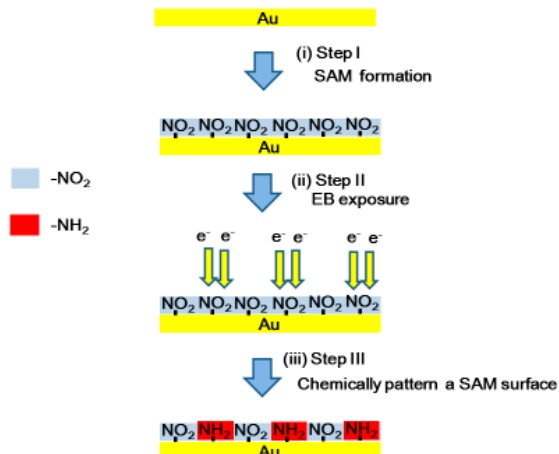

**Figure 2.** A scheme of the experimental steps. (Step I) SAM formation, (Step II) Patterning of SAM with EB lithography, and (Step III) EB patterning of a SAM, which induces transition of the SAM terminal functional aromatic $NO_2$ moieties to aromatic $NH_2$ moieties.

Patterning of SAMs was achieved by EB lithography for all features with various periods. SAMs were patterned at doses of 50 mC/cm$^2$. The exposed electron dose in this study is similar to the dose of 35,000 µC/cm$^2$ required to carry out the reduction of NO$_2$ on biphenyl based NO$_2$ terminated SAM on gold [51].

After EB pattering of SAM, PS-*b*-PMMA block copolymer thin films were formed onto the patterned SAM layer coated substrates. Next, they were annealed in a vacuum at 190 °C and was treated with an oxygen RIE to selectively remove the PMMA domains on the substrate. Figure 3 shows SEM micrographs of PMMA etched lamellar PS-*b*-PMMA block copolymer on Au-coated silicon substrates pre-coated with SAM. Lamella orientation of PS-*b*-PMMA block copolymer was observed irradiated by EB after 24 h of annealing at 190 °C. The patterned underlying substrates provide neutral surface layers for lamella orientation of PS-b-PMMA domains. However, perfect DSA was not achieved. It is known that the PMMA block preferentially wets on SAMs that contain polar groups and PMMA block preferentially wets on NO$_2$ region before EB irradiation. On the other hand, the NO$_2$ groups of the NPHT was reduced when EB was irradiated and were converted to NH$_2$ groups. Therefore, the surface energy of the underlying stripe pattern, which was composed of NO$_2$ and NH$_2$, was changed and became neutral layers for PS-b-PMMA. We observed the lamella orientation of PS-b-PMMA, whose dimension is approximately 30 nm. This dimension corresponds to lamella natural period of approximately 30 nm. We have also shown the EB irradiation control of surface energy of the patterned substrate. We could not observe the lamella orientation of PS-b-PMMA block copolymer at the dose of 10 mC/cm$^2$. The reason why perfect DSA could not be observed is thought to be for the following two reasons. First was the control of stripe width. It has been reported that the number of defects increased as the density multiplication factor, and the proportion of the chemical pattern period to the block copolymer period, increased [28]. Also, it is known that if the periods of the surface and block copolymer do not agree within approximately 10%, DSA of the block copolymer films will not be perfect [13]. The second reason is that it is difficult to modify with thiol monolayers on Au surfaces because our annealing temperature is 190 °C, which is much greater than the dissociation temperature of the Au-thiol bond. If we can reliably fabricate a chemically striped patterned substrate consisting of alternating stripes of SAM with an aromatic NO$_2$ terminal functionality and with aromatic NH$_2$ terminal functionality, which is a thermally stable SAM at a temperature of more than 190 °C, we may possibly accomplish perfect DSA.

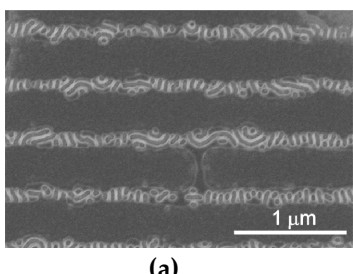 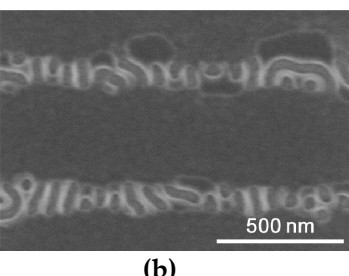 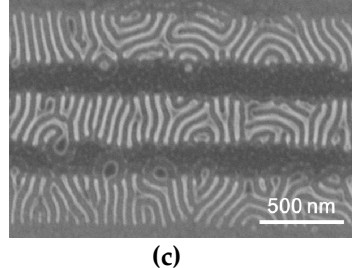

| (a) | (b) | (c) |

**Figure 3.** SEM micrographs of PMMA etched lamellar PS-*b*-PMMA block copolymer on Au deposited silicon substrates pre-coated with self-assembly monolayer (SAM). The widths of alternating stripes of SAMs were (**a**) 150 nm, (**b**) 150 nm and (**c**) 300 nm, respectively.

We investigated the annealing conditions in which the lammella orientation of PS-*b*-PMMA block copolymer was observed. Figure 4 shows SEM micrographs of PMMA etched lamellar PS-*b*-PMMA block copolymer on Au deposited silicon substrates pre-coated with SAM at the annealing temperature of (a) 150 °C and 250 °C, respectively. In both cases, we could not observe the lamella orientation of PS-*b*-PMMA block copolymer. This was because, while the self-assembly of PS-*b*-PMMA did not occur at the annealing temperature of 150 °C, the dissociation temperature of the Au-thiol bond would occur at the annealing temperature of 250 °C. These results indicated the vertical orientation conditions (perpendicular to the surface) required the appropriate annealing temperature. To our knowledge, this

is the first case of a neutral layer taking advantage of the reduction of SAM by EB irradiation, and this study provides insights on tuning the neutral layer and the vertical orientation conditions for suitable annealing temperature and irradiation doses. The main barrier for this process is to control the alignment of the domains to achieve perfect DSA. It is crucial to understand the process variables that control this alignment. Further work is in progress to fully characterize and elucidate all the mechanisms at play during the fabrication procedure and understand in detail the key $NO_2$ to $NH_2$ transition threshold, which are required to control the alignment of the domains in order to give DSA.

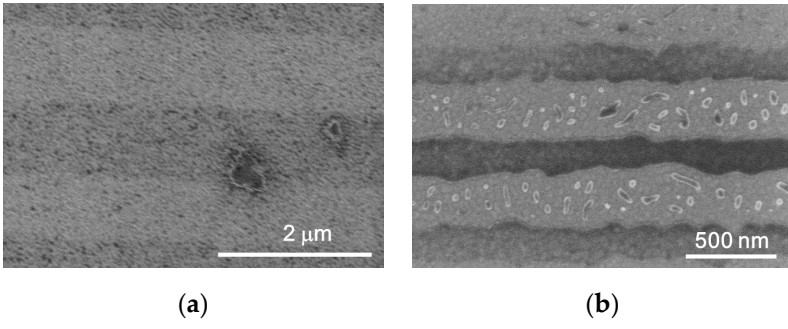

(**a**)　　　　　　　　　　　　　　　　　　　（**b**)

**Figure 4.** SEM micrographs of PMMA etched lamellar PS-*b*-PMMA block copolymer on Au deposited silicon substrates pre-coated with SAM at the annealing temperature of (**a**) 150 °C and (**b**) 250 °C, respectively.

Additionally, we created pre-patterned Si substrates by EB lithographic patterning and reactive ion etching. Our research has focused on self-aligned self-assembly of PS-*b*-PMMA block copolymer thin film patterns of line and space patterns and compared this with our simple DSA process. Figure 5 shows SEM micrographs of PMMA etched PS-*b*-PMMA block copolymer patterns on silicon substrates pre-coated with PS-*b*-PMMA. Line widths on these topographically patterned substrates were (a) 300 nm, (b) 100 nm, and (c) 300 nm, respectively. We successfully fabricated the DSA of PS-*b*-PMMA block copolymer onto pre-patterned surfaces resulting in subdivision of the lithographic patterns into line and space patterns. The spaces shown in the SEM images in Figure 5 correspond to perpendicular PMMA domain orientation within various trenches, and clearly demonstrates that perpendicular orientation of PS-*b*-PMMA was achieved. It can be observed that pre-patterned Si substrates induced excellent microphase segregation within the trenches regardless of the groove width with lamella domains orientating perpendicular to the substrate surface. However, these patterns lacked long-range alignment. Although the SEM micrograph is not shown here, sub-30 nm feature sizes and other patterns of PS-*b*-PMMA could be observed.

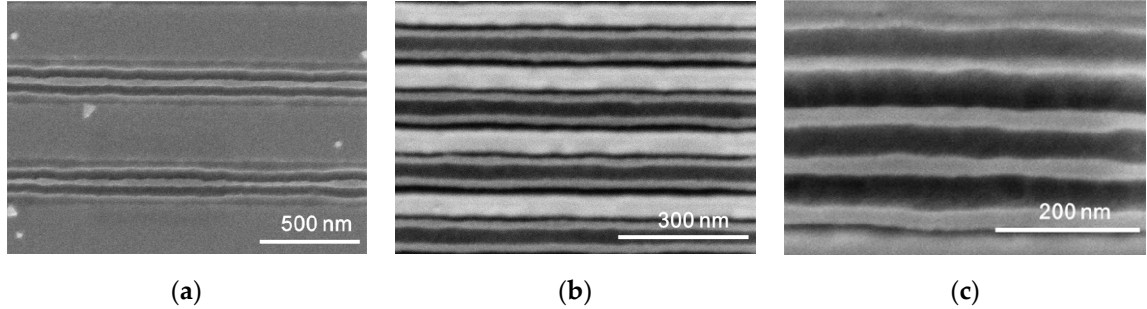

(**a**)　　　　　　　　　　　（**b**)　　　　　　　　　　　（**c**)

**Figure 5.** SEM micrographs of PMMA etched lamellar PS-*b*-PMMA block copolymer on silicon substrates pre-coated with PS-*b*-PMMA. Line widths on these topographically patterned substrates were (**a**) 300 nm, (**b**) 100 nm, and (**c**) 300 nm, respectively.

We have demonstrated a new method for the lamellar assembly of block copolymers by chemically patterned underlying substrates. We confirmed a process that uses directly EB patternable interfaces

using the chemical reduction of SAMs to decrease the current DSA process and avoid additional processes. In this work, the direct method has been clarified to be effective in inducing microphase separation such as lamella in block copolymer thin film on striped patterns of SAMs, but further work is in progress to optimize the process. Experiments exploiting novel SAM materials and block copolymers are ongoing in order to accomplish directed self-assembly by this method. Generally, the silane or silicon interface is stronger than the thiol or gold interface. Moreover, patterning on $SiO_2$ wafers is more compatible with top-down lithography. Additionally, the irradiation dose required for the patterning on $SiO_2$ is much lower than those required for a SAM on Au. It has been reported that the difference in EB lithography behavior between SAMs on Au and Si is related to the differing film thicknesses and electron scattering characteristics of the two underlying surfaces [54,55]. Therefore, the reduction of SAMs on $SiO_2$ substrate by EB and following DSA are in progress and they will be the subject of future reports.

## 4. Conclusions

We conclude that lamellar orientation of PS-*b*-PMMA block copolymer could be achieved using the transition of the $NO_2$ group to an $NH_2$ group induced by EB irradiation. The reduction of SAM substrates plays a crucial role in the ordering. By investing the lamellae orientation by EB, we found that the vertical state leads only to a suitable annealing temperature and irradiation dose. Future applications may be required to develop a new synthesis of sensitive and chemically specific underlying substrates, which change functionality directly upon EB irradiation. Our method has the potential to significantly simplify their processes as compared to currently used DSA process.

**Author Contributions:** Conceptualization, H.Y. and A.P.G.R.; methodology, H.Y.; formal analysis, H.Y.; investigation, H.Y.; sample preparation, H.Y. and G.D.; writing—original draft preparation, H.Y.; writing—review and editing, H.Y. and A.P.G.R.; project administration, H.Y.; funding acquisition, H.Y. discussion, T.K. All authors have read and agreed to the published version of the manuscript.

**Funding:** This work was supported in part by a Grant-in-Aid for Scientific Research (Project No. 16K14439) from the Ministry of Education, Culture, Sports, Science and Technology of Japan (MEXT).

**Acknowledgments:** This work was partly performed under the Cooperative Research Program of "Network Joint Research Center for Materials and Devices. Also, this work was partly performed under the Research Program for CORE lab of "Five-star Alliance" in "NJRC Mater. & Dev." A part of this work was supported by "Nanotechnorogy Platform Project (Nanotechnology Open Facilities in Osaka University)" of Ministry of Education, Culture, Sports, Science and Technology, Japan.

**Conflicts of Interest:** The authors declare no conflict of interest. The funders had no role in the design of the study; in the collection, analyses, or interpretation of data; in the writing of the manuscript, or in the decision to publish the results.

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
