# Peer review of "Lamellar Orientation of a Block Copolymer via an Electron-Beam Induced Polarity Switch in a Nitrophenyl Self-Assembled Monolayer or Si Etching Treatments"

_qubs, doi:10.3390/qubs4020019_

Round 1

Reviewer 1 Report

see attached

Author Response

We thank the reviewer for invaluable and thoughtful recommendations and advices.

We now revised the manuscript throughout according to your comment and advices.

1) Chemical analysis of the substrates pre and post modification to ensure the chemical changes proposed. This should be a straight forward measurement that is necessary in terms of the chemical nature of the patterning.

Unfortunately, we could not do experiment about chemical analysis of substrate pre and post modification because sample is limited due to international collaboration work and EB exposure tool’s machine time and analytical tools were limited at that time. For EB, it takes many time for large area because exposure dose is required for 50 mC/cm2 and XPS measurement is required for 1×1 cm2 area and we didn’t have the micro-XPS tool. This reason is why we couldn’t the data for chemical analysis. On the other hand, we fortunately could known XPS data of same SAM substrates by pre-and post- X-ray modification according to ref. 52 (Langmuir 24 (2008) 13969). Also, we could known XPS data of different SAM substrates by post- EB modification according to Ref.49 (Langmuir 20 (2004) 3766). Though we cannot know that these figures are corresponding to your recommendation, we can indirectly show the chemical analysis about the substrates by pre-and post- X-ray modification using X-ray (Figure 3 in Langmuir 24 (2008) 13969) and different SAM substrates by post- EB modification using EB (Figure1 in Langmuir 20 (2004) 3766). 

Response: We added below sentence.

Chemical analysis of substrate pre and post modification was not performed as EB exposure tool’s time and analytical tools were limited. XPS data of the SAM substrates by pre-and post- X-ray modification52 and XPS data of different SAM substrates by post- EB modification49 are available.

2) More in depth description of the phase segregation observed in figure 3.

Response: We changed below sentence.

After EB pattering of SAM, PS-b-PMMA block copolymer was spin-coated onto the patterned SAM layer coated substrates. Subsequently, they were annealed at 190 ËšC, in a vacuum and was treated with an oxygen RIE process in order to selectively remove the PMMA domains and form PS patterns on the substrate. Figure 3 shows SEM images of PMMA etched lamellar PS-b-PMMA block copolymer on Au coated silicon substrates pre-coated with SAM. Lamella orientation of PS-b-PMMA block copolymer was observed irradiated by EB after 24h of annealing at 190 ËšC. The resulting chemical patterns provide neutral surface layers for lamella orientation of PS-b-PMMA domains. However, perfect DSA was not achieved. It is known that the PMMA block preferentially wets on SAMs that contain polar groups and PMMA block preferentially wets on NO2 region before EB irradiation. On the other hand, the surface NO2 groups of the NPHT become reduced upon EB exposure and are converted to NH2 groups. Therefore, the surface energy of stripe pattern underlayer, which is composed of NO2 and NH2, is changed and becomes neutral layers for PS-b-PMMA. We observed lamella orientation of PS-b-PMMA, whose dimension is c.a. 30 nm. This dimension corresponds to lamella natural period of approximately 30 nm. We have also shown, EB irradiation control of surface energy of the patterned substrate. We couldn’t observe lamella orientation of PS-b-PMMA block copolymer at the dose of 10 mC/cm2. The reason why perfect DSA couldn’t be observed is thought to be for two reasons. First was the control of stripe width. It has been reported that the number of defects increased as the density multiplication factor, and the ratio of the chemical pattern period to the block copolymer period, increased [55]. Also, it is known that if the periods of the surface and block copolymer don’t agree within approximately 10%, the morphology of the block copolymer films will not be perfect [13]. The second reason is that Au surfaces are difficult to modify with thiol monolayers because our annealing temperatures are 190 ËšC, which is much greater than the dissociation temperature of the Au-thiol bond. If we can reliably fabricate chemically striped patterned substrate consisting of alternating stripes of SAM with an aromatic NO2 terminal functionality and with aromatic NH2 terminal functionality, which is thermally stable SAM at the temperature of more than 190 ËšC, we have possibility to accomplish perfect DSA.

3) Of recommendation to the authors but not necessary for publication would be: • AFM of the materials to better understand the surface roughness of the system and how this may also be impacting observed results, particularly at boundaries. Other techniques may also be useful however AFM could provide appropriately scaled information as to the polymer organization but more importantly the boundaries and interfaces. Phase imaging could also be useful in regards to the polymer segregation effects.

We investigated the phase image of block copolymer by AFM measurement at first because I don’t know know-how to observe the phase-separation of block copolymer.  We studied different composition of block copolymer such as cylinder as preliminary experiment. As you can see, we couldn’t get clear AFM image as shown in above AFM image at my first study of block copolymer. Therefore, we couldn’t get polymer segregation effects as you advice. For this reason, we tried to observe SEM image after O2 plasma etching. As you can see, we could obtain the clear SEM image as shown in above SEM image and we adopt SEM measurement in observation of phase separation of block copolymer.

Response: We corrected.

We believe the revised manuscript is now suitable for publication in Quantum Beams Appling to Innovative Industrial Materials.

                                             Sincerely Yours

                                             Hiroki Yamamoto

Reviewer 2 Report

The authors report on directed self-assembly of block-copolymers by pre-structuring of the substrate. Some issues should be clarified prior to publication:

  • Which PS-b-PMMA was used (Polymer-Source offers more than 50)?
  • The authors claim that the SAM modification induces lamellar orientation of the SAM. However, the presentation of experiments does not support the manuscript title and abstract. The "appropriate irradiation dose and annealing temperature" SEM image is not shown (that should have been the 190°C one).
  • Only on RIE SiO2 the treatment was proven to be successfull, however, this experimental finding is neither reflected in abstract nor title.

Minor:

  • the abbreviation DSA is only explained the second time it is used
  • as piranha solution is a highly dangerous chemical, I recommend to add a warning in brackets to the experimental section
  • in Fgure 1, it would be good to assign the shown steps' numbers to the procedure mentioned in the text (e.g. develop (iii) )

Author Response

We now largely revised the manuscript throughout according to your comment.

We thank the reviewer for invaluable and thoughtful comments.

1) Which PS-b-PMMA was used (Polymer-Source offers more than 50)?

Response: We modified as below.

Lamella forming PS-b-PMMA (PS: 53kg/mol, PMMA: 54kg/mol) was purchased from Polymer Source, Inc and used as received.

2) The authors claim that the SAM modification induces lamellar orientation of the SAM. However, the presentation of experiments does not support the manuscript title and abstract. The "appropriate irradiation dose and annealing temperature" SEM image is not shown (that should have been the 190°C one).

Response: We accounted for your comment as below and added below sentence.

Thank you very much for appropriate comment for our study. We showed that the SEM images of 190°C one in Figure 3. We unfortunately could not present perfect directed self-assembly by our method as your comment. It is because Au-S bonds of SAMs used in this study maybe was partially decomposed at that temperature. Therefore, we try to report that we change the SAM or block copolymer and we will be in progress to obtain perfect self-assembl. However, in this article we could not perfect DSA by this SAM, but we observed appropriate an irradiation dose and annealing temperature is required for lamella orientation of PS-b-PMMA in local exposed area as shown in Figure 3. We could not do experiment about perfect DSA of substrate because SAM sample may be partially unstable for 190°C and they were limited to use due to international co-laboration work. Also, EB exposure tool’s machine time was limited. Though we can understand your comment, we would like to publish this in present state as first step for the reason for above. 

Response: We changed below sentence.

After EB pattering of SAM, PS-b-PMMA block copolymer was spin-coated onto the patterned SAM layer coated substrates. Subsequently, they were annealed at 190 ËšC, in a vacuum and was treated with an oxygen RIE process in order to selectively remove the PMMA domains and form PS patterns on the substrate. Figure 3 shows SEM images of PMMA etched lamellar PS-b-PMMA block copolymer on Au coated silicon substrates pre-coated with SAM. Lamella orientation of PS-b-PMMA block copolymer was observed irradiated by EB after 24h of annealing at 190 ËšC. The resulting chemical patterns provide neutral surface layers for lamella orientation of PS-b-PMMA domains. However, perfect DSA was not achieved. It is known that the PMMA block preferentially wets on SAMs that contain polar groups and PMMA block preferentially wets on NO2 region before EB irradiation. On the other hand, the surface NO2 groups of the NPHT become reduced upon EB exposure and are converted to NH2 groups. Therefore, the surface energy of stripe pattern underlayer, which is composed of NO2 and NH2, is changed and becomes neutral layers for PS-b-PMMA. We observed lamella orientation of PS-b-PMMA, whose dimension is c.a. 30 nm. This dimension corresponds to lamella natural period of approximately 30 nm. We have also shown, EB irradiation control of surface energy of the patterned substrate. We couldn’t observe lamella orientation of PS-b-PMMA block copolymer at the dose of 10 mC/cm2. The reason why perfect DSA couldn’t be observed is thought to be for two reasons. First was the control of stripe width. It has been reported that the number of defects increased as the density multiplication factor, and the ratio of the chemical pattern period to the block copolymer period, increased [55]. Also, it is known that if the periods of the surface and block copolymer don’t agree within approximately 10%, the morphology of the block copolymer films will not be perfect [13]. The second reason is that Au surfaces are difficult to modify with thiol monolayers because our annealing temperatures are 190 ËšC, which is much greater than the dissociation temperature of the Au-thiol bond. If we can reliably fabricate chemically striped patterned substrate consisting of alternating stripes of SAM with an aromatic NO2 terminal functionality and with aromatic NH2 terminal functionality, which is thermally stable SAM at the temperature of more than 190 ËšC, we have possibility to accomplish perfect DSA.

3) Only on RIE SiO2 the treatment was proven to be successfull, however, this experimental finding is neither reflected in abstract nor title.

Response: We reflected the title and abstract.

We changed the title as below.

Lamellar orientation of a block copolymer via an electron-beam induced polarity switch in a nitrophenyl self-assembled monolayer or Si etching treatments.

We added the sentence in abstract as below according to your kind comment.

We also fabricated surface topography by EB lithographic patterning and reactive ion etching (RIE) of silicon. DSA onto such pre-patterned Si substrates was proven to be successful for subdivision of the lithographic patterns into line and space patterns.

4) the abbreviation DSA is only explained  the second time it is used

Response: We corrected.

We spelled out abbreviations after first occurrence, L37.

In particular, directed self-assembly (DSA) of block copolymers has attracted significant attention ~ (L37)

Also, we deleted the spelling out of second occurrence DSA as below.

Directed self-assembly (DSA) is currently being ~ ⇒ DSA is currently being~ (L40).

5) as piranha solution is a highly dangerous chemical, I recommend to add a warning in brackets to the experimental section

Response: We added below sentence.

(Warning: Piranha solution is very reactive and corrosive)

6) in Figure 1, it would be good to assign the shown steps' numbers to the procedure mentioned in the text (e.g. develop (iii)

Response: We corrected in Figure 1 caption below.

Figure 1. (a) Schematic presentation of block copolymer lamella orientation process using a conventional chemically patterned surface. (i) SAM coating on Si. (ii) resist coating. (iii) Resist was exposed and developed. (iv) SAM was patterned in the presence of O2 though resist mask (v) Resist was removed and surface nutrality. (vi) Block copolymer was spin-coated. (vii) Block copolymer was self-assembled to match underlying SAM patterns. (b) Schematic presentation of block copolymer lamella orientation process using a directly chemically patternable SAMs method. (i) SAM coating on Si. (ii) SAM was directly patterned by EB. (iii) Block copolymer was spin-coated. (iv) Lammella orientation of block copolymer was self-assembled in underlying SAM patterns.

We believe the revised manuscript is now suitable for publication in Quantum Beams Appling to Innovative Industrial Materials.

                                             Sincerely Yours

                                             Hiroki Yamamoto

Reviewer 3 Report

The authors demonstrate a novel method for directed self-assembly of PS-PMMA block copolymers on Au coated Si surfaces. The DSE is carried out using e-beam modification of a self assembled monolayer of NO2, which upon e-beam irradiation reduces to NH2. The paper deserves publication after a minor revision.

  • Spell out abbreviations after first occurrence e.g. DSE, L37
  • Figure and captions must be self-explanatory. E.g. in figure 1, explain process sequences indicated by Roman numbers.
  • L96: claim that "hexagonally packed hole arrays" demonstrated not supported by data
  • L114. Oxygen plasma is not generally selective for PMMA. It will remove also the PS blocks (see e.g. Telecka RSC Adv., 2018, 8, 4204–4213). Authors are requested to verify/explain this claim, by e.g. providing data for pattern morphology vs oxygen-plasma exposure time or power.  
  • L159-: the explanation of figure 3 is not very clear. It is unclear which message the authors try yo convey here. Are the authors trying to explain a problem or why it works?
  • L180: What is meant by "vertical orientation conditions"? Is it perpendicular to the surface, or perpendicular to the e-beam defined lines?

Author Response

We now largely revised the manuscript throughout according to your comment.

We thank the reviewer for invaluable and thoughtful comments.

1) Spell out abbreviations after first occurrence e.g. DSE, L37

Response: We corrected.

We spelled out abbreviations after first occurrence, L37.

In particular, directed self-assembly (DSA) of block copolymers has attracted significant attention ~ (L37)

Also, we deleted the spelling out of second occurrence DSA as below.

Directed self-assembly (DSA) is currently being ~ ⇒ DSA is currently being~ (L40).

2) Figure and captions must be self-explanatory. E.g. in figure 1, explain process sequences indicated by Roman numbers.

Response: We corrected in Figure 1 caption below.

Figure 1. (a) Schematic presentation of block copolymer lamella orientation process using  a conventional chemically patterned surface. (i) SAM coating on Si. (ii) resist coating. (iii) Resist was exposed and developed. (iv) SAM was patterned in the presence of O2 though resist mask (v) Resist was removed and surface nutrality. (vi) Block copolymer was spin-coated. (vii) Block copolymer was self-assembled to match underlying SAM patterns. (b) Schematic presentation of block copolymer lamella orientation process using a directly chemically patternable SAMs method. (i) SAM coating on Si. (ii) SAM was directly patterned by EB. (iii) Block copolymer was spin-coated. (iv) Lammella orientation of block copolymer was self-assembled in underlying SAM patterns.

3) L96: claim that "hexagonally packed hole arrays" demonstrated not supported by data

Response: We corrected.

We deleted the sentence of or hexagonally packed hole arrays as below.

~ line and space patterns or hexagonally packed hole arrays. (L96)

4) L114. Oxygen plasma is not generally selective for PMMA. It will remove also the PS blocks (see e.g. Telecka RSC Adv., 2018, 8, 4204–4213). Authors are requested to verify/explain this claim, by e.g. providing data for pattern morphology vs oxygen-plasma exposure time or power.  

Response: We accounted for your comment as below.

As you mentioned, oxygen plasma induced etching of both polystyrene (PS) and PMMA. However, the etching durability is different. I don’t have the data about etching of PS, but I have the data of polyhydroxy styrene (PHS). We can provide the data for pattern morphology vs oxygen-plasma exposure time in PMMA and PHS. The etching durability of PS is higher than that of PHS. From above Figure, the etching durability of PHS is two-fold higher than that of PMMA. Therefore, the etching durability of PS is more than two-fold higher compared to PMMA. Therefore, we could obtain SEM images of remained PS pattern in Figure 3 and Figure 5.

5) L159-: the explanation of figure 3 is not very clear. It is unclear which message the authors try to convey here. Are the authors trying to explain a problem or why it works?

Response: We changed below sentence.

After EB pattering of SAM, PS-b-PMMA block copolymer was spin-coated onto the patterned SAM layer coated substrates. Subsequently, they were annealed at 190 ËšC, in a vacuum and was treated with an oxygen RIE process in order to selectively remove the PMMA domains and form PS patterns on the substrate. Figure 3 shows SEM images of PMMA etched lamellar PS-b-PMMA block copolymer on Au coated silicon substrates pre-coated with SAM. Lamella orientation of PS-b-PMMA block copolymer was observed irradiated by EB after 24h of annealing at 190 ËšC. The resulting chemical patterns provide neutral surface layers for lamella orientation of PS-b-PMMA domains. However, perfect DSA was not achieved. It is known that the PMMA block preferentially wets on SAMs that contain polar groups and PMMA block preferentially wets on NO2 region before EB irradiation. On the other hand, the surface NO2 groups of the NPHT become reduced upon EB exposure and are converted to NH2 groups. Therefore, the surface energy of stripe pattern underlayer, which is composed of NO2 and NH2, is changed and becomes neutral layers for PS-b-PMMA. We observed lamella orientation of PS-b-PMMA, whose dimension is c.a. 30 nm. This dimension corresponds to lamella natural period of approximately 30 nm. We have also shown, EB irradiation control of surface energy of the patterned substrate. We couldn’t observe lamella orientation of PS-b-PMMA block copolymer at the dose of 10 mC/cm2. The reason why perfect DSA couldn’t be observed is thought to be for two reasons. First was the control of stripe width. It has been reported that the number of defects increased as the density multiplication factor, and the ratio of the chemical pattern period to the block copolymer period, increased [55]. Also, it is known that if the periods of the surface and block copolymer don’t agree within approximately 10%, the morphology of the block copolymer films will not be perfect [13]. The second reason is that Au surfaces are difficult to modify with thiol monolayers because our annealing temperatures are 190 ËšC, which is much greater than the dissociation temperature of the Au-thiol bond. If we can reliably fabricate chemically striped patterned substrate consisting of alternating stripes of SAM with an aromatic NO2 terminal functionality and with aromatic NH2 terminal functionality, which is thermally stable SAM at the temperature of more than 190 ËšC, we have possibility to accomplish perfect DSA.

6) L180: What is meant by "vertical orientation conditions"? Is it perpendicular to the surface, or perpendicular to the e-beam defined lines?

Response: We modified as below.

vertical orientation conditions  ⇒vertical orientation conditions (perpendicular to the surface)

We believe the revised manuscript is now suitable for publication in Quantum Beams Appling to Innovative Industrial Materials.

                                             Sincerely Yours

                                             Hiroki Yamamoto

Round 2

Reviewer 2 Report

The authors managed to improve the manuscript so that it can be followed more easily and title and abstract do not promise more than the manuscript can provide anymore. The article can be accepted in the current form.